# It Hears, It Sees too: Multi-Modal LLM for Depression Detection By Integrating Visual Understanding into Audio Language Models

## Abstract

Depression is one of the most prevalent mental health disorders globally. In recent years, multi-modal data, such as speech, video, and transcripts, has been increasingly used to develop AI-assisted depression assessment systems. Large language models have further advanced this field due to their strong language understanding and generalization capabilities. However, conventional LLMs remain text-centric and cannot process the rich non-verbal cues found in audio and visual modalities, which are critical components in mental health evaluation. While multi-modal LLMs offer a promising direction, few are tailored for psychological applications. In this study, we propose a novel multi-modal LLM framework for depression detection. Our approach augments an audio language model with visual understanding and aligns audio-visual features at the timestamp level. This fine-grained alignment improves modeling of temporal dynamics across modalities while reducing the need for extensive training data and computational resources. Experiments on the DAIC-WoZ dataset demonstrate that our model outperforms both single-modality approaches and previous multi-modal methods. Moreover, the proposed framework can be extended to incorporate additional physiological signals, paving the way for broader clinical applications beyond mental health.

## 1 Introduction

Depression has emerged as a critical concern in the field of mental health, affecting a broad population across various age groups. Particularly, the incidence of depression among adolescents has surged over the past decade, raising significant social and public health concerns (Thapar et al., 2022). Diagnosing and treating depression often entails substantial labor and financial costs for both families and healthcare systems. With the advancement of natural language processing (NLP), increasing attention has been given to automated approaches for depression detection, reducing human intervention. Large language models (LLMs) have demonstrated remarkable capabilities across a wide array of NLP tasks (Naveed et al., 2023), which has sparked interest in their application to mental health screening (Hengle et al., 2024; Xu et al., 2024). Despite their success, a fundamental limitation of conventional LLMs lies in their confinement to textual inputs, lacking the capacity to interpret multi-modal signals such as speech and facial expressions that are also indicative of depressive symptoms (Koops et al., 2023; Krause et al., 2021).

Multi-modal data, including acoustic and visual cues, can significantly enhance the accuracy of depression detection. Prior studies have shown that individuals at high risk of depression often exhibit reduced facial expressiveness, diminished vitality, and weakened responses to external stimuli such as decreased eye contact (Perez & Riggio, 2003; Waxer, 1974). Similarly, specific acoustic features, such as monotonous tone, slow speech rate, disfluency, and low vocal energy, have been linked to depressive states (Koops et al., 2023). These behavioral signals offer valuable complementary information beyond what can be derived from text alone. Multi-modal large language models (MLLMs) offer an ideal solution to the integration of text and multi-modal data, which shows great promise in a lot of downstream tasks (Zhang et al., 2024a). However, current MLLMs face several limitations that hinder their application to depression detection. First, depression detection relies heavily on temporal data such as audio and video, yet most existing MLLMs are limited to static images (Caffagni et al., 2024). Furthermore, due to the relatively small size of depression-related datasets compared to

Figure 1: The training scheme of the proposed multi-modal LLM for depression detection.

standard NLP corpora, developing MLLMs for this domain demands careful consideration of model complexity to mitigate overfitting and ensure training efficiency.

To address these limitations, we propose a simple yet effective framework that adapts a multi-modal large language model for depression detection. Our method builds upon a pretrained audio language model (ALM) and augments it with visual understanding capabilities, forming a truly multi-modal system. This design leverages the shared temporal structure of audio and visual modalities, allowing for the alignment at the timestamp level. By incrementally integrating visual modules into the ALM with self-supervised visual pretraining and parameter-efficient fine-tuning (PEFT) (Hu et al., 2022), our approach maintains the efficiency and modularity of the base model while enhancing its multi-modal capacity. This strategy also reduces the number of trainable parameters and mitigates the need for large-scale pretraining, making it efficient in data usage and computational requirements. Experiments on the public depression detection dataset, DAIC-WoZ, confirm the effectiveness of our approach, highlighting its potential for practical applications in mental health assessment.

In summary, the contributions of this work consist of the following aspects:

- We develop a multi-modal large language model for depression detection based on the Qwen2-Audio (Chu et al., 2024) model by integrating a self-supervised vision encoder with parameter-efficient fine-tuning. To the best of our knowledge, this is the first study to propose **multi-modal depression detection using LLM across text, audio, and video modalities**;

- We implement a timestamp-level alignment strategy that enables fine-grained temporal fusion across modalities. This design leverages the inherent temporal characteristics of both audio and video signals, enhancing the model's capacity to capture subtle behavioral cues indicative of depression.

- We validate our approach by the comparison with single-modality methods and previous LLM-based state-of-the-art methods on the DAIC-WoZ database (Gratch et al., 2014). The experimental results demonstrate that our approach yields superior performance at a smaller model scale (7B versus 13B), compared with pioneering multi-modal LLMs.

## 2 RELATED WORKS

### 2.1 AUTOMATED DEPRESSION DETECTION

Deep learning has been widely adopted for automated depression detection using speech, text, and video modalities. Earlier works focused on single modality, such as self-supervised speech models (Wu et al., 2023), hierarchical acoustic representations (Chen et al., 2022), or mobile speech data (Kim et al., 2023). Visual features like facial expressions and eye movements have also shown promise, with methods leveraging weakly supervised learning (Shangguan et al., 2022), gaze patterns (Zheng et al., 2024), and combined facial-gaze analysis (Stolicyn et al., 2022). Recent studies have explored multi-modal fusion to capture richer cues, incorporating audio, video, and text (Zhang et al., 2024c; Shen et al., 2022; Xue et al., 2024). However, most rely on late fusion strategies without joint pretraining, limiting their ability to fully exploit temporal and semantic correlations across modalities.

## 2.2 LARGE LANGUAGE MODELS IN DEPRESSION

Large language models have been applied to depression detection due to their strong ability to model long-range dependencies in dialogue, which is an essential feature for analyzing clinical interviews. For example, Liu *et al.* (Liu et al., 2023b) introduced ChatCounselor, which leverages LLMs to assess depressive symptoms and provide mental health support. Other studies have employed LLMs to analyze social media content; Hengle *et al.* (Hengle et al., 2024) constructed a benchmark for depression-stress classification from online posts, while Xu *et al.* (Xu et al., 2024) used LLMs to infer depression status from various web-based sources. Recent efforts have extended LLMs to multi-modal settings for improved diagnostic accuracy. Sadeghi *et al.* (Sadeghi et al., 2024) combined LLMs with facial expression analysis to estimate depression severity, and Zhang *et al.* (Zhang et al., 2024b) incorporated acoustic landmarks into LLMs to build an audio-text model for depression detection. While these approaches demonstrate the potential of LLMs in mental health applications, they remain limited to textual inputs or approximations thereof (e.g., acoustic landmarks). The inability to directly process rich multi-modal signals restricts their overall effectiveness.

## 2.3 MULTI-MODAL LARGE LANGUAGE MODELS

Integrating textual inputs with audio and visual modalities represents a major advancement in the development of generative AI. The fusion of LLMs with visual encoders has enabled impressive performance on tasks such as visual dialogue, visual question answering, and image captioning (Liu et al., 2023a; Zhu et al., 2023; Dai et al., 2023; Wang et al., 2024; Lu et al., 2024). Similarly, audio language models have emerged to jointly process speech and text. For instance, Chu *et al.* (Chu et al., 2024) introduced Qwen2-Audio, extending the Qwen2-7B backbone (Qwen et al., 2025), while Ding *et al.* (Ding et al., 2025) proposed Kimi-Audio, which incorporates both discrete acoustic tokens and continuous audio embeddings into an LLM framework. Despite their success, these models are generally not well-suited for mental health applications due to substantial domain gaps in both training data and pretraining objectives. Moreover, most vision-language models lack the capacity to handle continuous video input, further limiting their applicability to tasks such as depression detection, where temporal visual cues are crucial.

# 3 METHOD

## 3.1 OVERVIEW OF THE FRAMEWORK

We propose a multi-modal large language model (MLLM) for depression detection, constructed upon a pretrained audio language model (ALM) as the backbone. As depicted in Figure 2, the framework consists of three key components: (1) an **audio encoder** that processes raw audio signals and extracts temporal embeddings; (2) a **visual encoder** that receives video frames and produces visual embeddings aligned with the audio stream at the timestamp level; (3) a **large language model** that integrates the audio-visual features along with textual inputs to perform depression classification.

The training process is divided into three sequential stages. First, the visual encoder is pretrained using a self-supervised learning strategy inspired by masked autoencoders (He et al., 2022), which enhances its capacity to capture rich visual representations. In the second stage, the visual encoder is fine-tuned on a contrastive alignment task designed to match visual and audio embeddings at the utterance level, thereby improving cross-modal temporal synchronization. Finally, the projection layer and LLM are trained using parameter-efficient fine-tuning (PEFT) techniques to effectively incorporate the visual modality while minimizing additional computational overhead.

## 3.2 MODEL COMPONENTS

### 3.2.1 AUDIO LANGUAGE MODEL

We adopt Qwen2-Audio (Chu et al., 2024) as the foundation of our framework. This model integrates Whisper-large-v3 (Radford et al., 2023) as the audio encoder and Qwen2-7B as the language model. The audio encoder processes raw waveforms resampled to 16 kHz and converts them into 128-channel Mel-spectrograms, with each frame representing a 10 ms segment. These spectrograms are subsequently downsampled via strided convolutions and average pooling, resulting in encoder

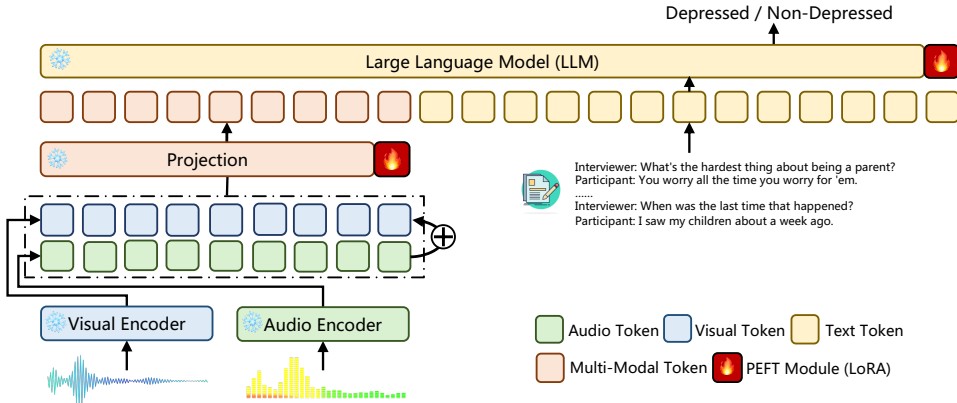

Figure 2: The framework of the proposed multi-modal large language model. The model includes an audio encoder, a visual encoder, and an LLM for detection.

outputs where each frame corresponds to a 40 ms segment of the original waveform. To ensure the universality of our method, we retain the pretrained weights of Qwen2-Audio throughout the initial stages and apply PEFT-based adaptation only in the final training phase. Notably, our framework is modular and can be extended to other audio language models, provided their audio encoders output sequences aligned with fixed temporal intervals.

### 3.2.2 VISUAL ENCODER

The visual encoder is designed to extract visual embeddings that align temporally with the audio encoder outputs. To ensure architectural compatibility and ease of alignment, its design mirrors the Whisper encoder, comprising a strided convolutional embedding layer, a stack of Transformer encoder layers, and an output average pooling layer. Initially, visual features are resampled to match the temporal resolution of the audio Mel-spectrograms and are projected into the embedding space via 1D convolutions. This embedding process includes striding, reducing the temporal resolution to 20 ms per token. The resulting features are then processed by the Transformer layers and further downsampled through average pooling to match the final 40 ms resolution of the audio encoder outputs. As a result, both audio and visual embeddings are temporally synchronized, as illustrated in Figure 3.

### 3.2.3 AUDIO-VISUAL PROJECTION

After obtaining audio and visual embeddings, the next step is to fuse them into a unified representation for input into the LLM. While a common fusion strategy involves concatenating modality embeddings along the sequence dimension (Xu et al., 2025), this approach is suboptimal for integrating new modalities into pretrained LLMs, as it disrupts the expected sequence length and can interfere with positional encoding. To preserve compatibility with pretrained LLMs, we propose a simple yet effective fusion method—element-wise addition of audio and visual embeddings, which is illustrated in Figure 2. This is feasible due to our explicit timestamp-level synchronization, ensuring both sequences share the same temporal structure. Moreover, our three-stage training strategy progressively aligns the modalities, enabling effective fusion without representation collapse.

### 3.3 TIMESTAMP-SYNCHRONIZED DATA AUGMENTATION

Depression corpora typically consist of participant–interviewer interviews, which present two challenges: (1) severe class imbalance, as healthy controls far outnumber depressed individuals, and (2) limited data volume, despite long session durations. To alleviate these issues, we adopt subdialogue shuffling based on Wu et al. (2023), segmenting lengthy interviews into shorter, contiguous exchanges. This increases sample size per participant and enables flexible resampling for class balancing.

Building on Wu et al. (2023), we enhance the method by ensuring timestamp alignment across transcript, audio, and visual modalities. Each subdialogue is constrained to start with an interviewer's

utterance and end with the participant's response, maintaining contextual coherence and narrowing the domain gap between LLM pretraining and depression detection. We then discard interviewer audio and corresponding visual frames, retaining only participant segments, while preserving interviewer transcripts. This choice reflects two considerations: interviewer speech carries little acoustic value for mental state assessment, yet their utterances are essential for conversational coherence. Although removing interviewer segments inevitably discards some multimodal information, the trade-off between information reduction and coherence is analyzed in Section 4.3.3. Further augmentation details are provided in the Appendix.

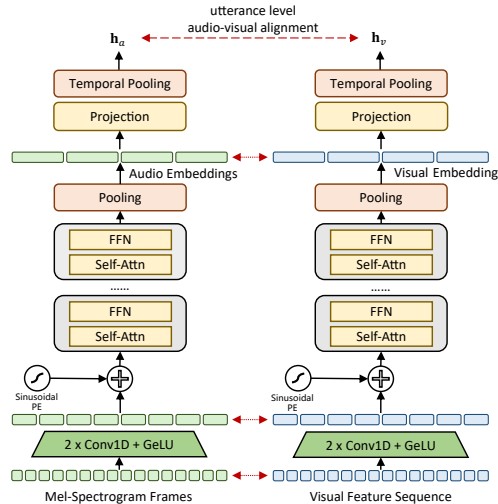

Figure 3: The scheme of utterance-level audio-visual alignment. Audio and visual inputs are strided simultaneously, ensuring synchronization on timestamps.

### 3.4 TRAINING

The training pipeline of our framework is divided into three sequential stages, as shown in Figure 1. The first two stages focus on training the visual encoder, while the final stage involves fine-tuning the LLM.

#### 3.4.1 SELF-SUPERVISED VISUAL PRETRAINING

To enhance the visual representation capability of the encoder, we first conduct self-supervised pretraining. Instead of learning directly from raw video data, we opt to pretrain on pre-extracted visual features, as raw video files may contain sensitive content and are often unavailable in commonly used depression-related corpora. This not only addresses potential privacy concerns but also reduces computational overhead, making the approach more generalizable to other time-series modalities such as physiological signals (e.g., rPPG and ECG).

Inspired by the masked autoencoder (MAE) framework (He et al., 2022), we design a reconstruction task where the encoder learns to recover masked portions of the input time series. Specifically, given a sequence input $\mathbf{x} = (x_1, x_2, ..., x_T) \in \mathbb{R}^{T \times d}$, we randomly mask $K$ frames of the input and use a learnable token $x_{mask} \in \mathbb{R}^d$ shared across all masked frames. The indices for masked tokens are denoted as $\mathcal{M}$, and the indices for unmasked tokens are denoted as $\mathcal{V}$. Obviously $\mathcal{V} \cup \mathcal{M} = \{1, 2, \ldots, T\}$. The unmasked sequence $\mathbf{x}_{in} = \{x_i | i \in \mathcal{V}\} \in \mathbb{R}^{K \times d}$ are fed to the visual encoder to acquire the latent representation $\mathbf{h} \in \mathbb{R}^{T-K \times d}$. Then, the latent representation $\mathbf{h}$ and masked frames are concatenated together to acquire the input sequence $\mathbf{z} \in \mathbb{R}^{T \times d}$, which are fed to the decoder to obtain the reconstructed input sequence $\hat{\mathbf{x}} \in \mathbb{R}^{T \times d}$. The objective is to minimize the mean squared error (MSE) between the reconstructed and original sequences within the masked regions:

$$\min \frac{1}{|\mathcal{M}|} \sum_{i \in \mathcal{M}} ||\hat{x}_i - x_i||_2^2 \tag{1}$$

This approach allows the model to capture temporal dependencies and improve robustness in downstream tasks.

#### 3.4.2 UTTERANCE LEVEL AUDIO-VISUAL ALIGNMENT

After the visual pretraining in the first stage, the visual encoder is enabled to extract visual embeddings from input visual feature sequences for downstream tasks. However, the visual comprehension of the visual encoder is not aligned with the audio encoder. To reduce the training gap between both encoders, we design a proxy downstream task with contrastive learning to align the visual encoder with the audio encoder at the utterance level. As illustrated in Figure 3, we add a projection layer to each encoder, respectively, and pool the outputs in the time dimension to obtain the utterance level

representations. Given a mini-batch of audio outputs $\mathbf{h}_a \in \mathbb{R}^{N \times d}$ and visual outputs $\mathbf{h}_v \in \mathbb{R}^{N \times d}$, where $N$ denotes the batch size, we obtain a similarity matrix $\mathbf{Sim} = \mathbf{h}_a \mathbf{h}_v^T \in \mathbb{R}^{N \times N}$. The learning objective is to find the correct match of each audio-visual pair for utterance level audio-visual alignment:

$$\min \mathcal{L}_{ce}(\mathbf{Sim}/tau, \mathbf{I}_N) \tag{2}$$

where $\mathcal{L}_{ce}$ denotes cross-entropy loss, $\mathbf{I}_N$ denotes the identity matrix, and $\tau$ is the temperature parameter.

During this stage, we freeze the entire audio encoder and the lower layers of the visual encoder to preserve the representations learned in the initial stage. Only the upper layers of the visual encoder receive gradient updates, ensuring stability and preventing catastrophic forgetting.

### 3.4.3 Multi-Modal Instruction Tuning

In the final stage, we integrate the pretrained visual encoder with the audio language model to construct a multi-modal large language model tailored for depression detection, which is illustrated in Figure 2. Since traditional LLMs are not inherently designed to process visual information, additional instruction tuning is required to adapt the model to this task. We employ Low-Rank Adaptation (LoRA) (Hu et al., 2022) to update the parameters of both the LLM and the modality projection layer. As audio and visual features have been temporally synchronized and aligned at the utterance level in previous stages, the complexity of cross-modal fusion is substantially reduced.

### 3.5 Multi-Scale Sliding-Window Inference

Since our model is trained on subdialogues rather than entire conversations, we adopt a multi-scale sliding-window inference strategy to derive a final prediction for each full conversation. This approach aggregates predictions from multiple subdialogue segments extracted at different temporal scales. Specifically, for each conversation, we generate a fixed number (200) of subdialogues at three predefined durations: 30s, 75s, and 120s. This multi-scale design ensures that each temporal resolution contributes equally to the final decision, capturing both short-term and long-term behavioral cues. The overlap between adjacent subdialogues is dynamically adjusted based on the conversation length and the total number of segments per setting. Each time-scale configuration yields an independent conversation-level prediction, and the final prediction is determined by majority voting across the three settings.

## 4 Experiments

### 4.1 Database and Implementation Details

We utilize the DAIC-WoZ database (Gratch et al., 2014), one of the most popular datasets for depression detection, to develop and evaluate our proposed multi-modal LLM in depression detection. The DAIC-WoZ database contains interview transcripts, speech records, and visual features from 189 participants, including healthy controls and depression cases. The golden labels of the dataset are based on PHQ-8 scores, where a PHQ-8 score higher than 10 is recognized as a depressed case. The training set contains 107 participants, 30 of whom are labeled as depressed, while the development set contains 35 participants, 12 of whom are labeled as depressed. Following our previous works (Wu et al., 2023; Zhang et al., 2024b), we report the evaluation results on the development set for comparison. In addition to the training set and development set, we also evaluated our method on the test set, where 14 out of the 47 subjects are labeled as depressed. For timestamp-synchronized data augmentation, we set the maximum length of each subdialogue to 120 seconds, generate 1,000 subdialogues per conversation with depression, which achieves a trade-off between data diversity and the risk of overfitting. The visual features generated by data augmentation are utilized for self-supervised visual pretraining and utterance level audio-visual alignment. Then the augmented transcripts, audio clips, and visual features are used for multi-modal instruction finetuning. Our multi-modal LLM for depression detection is developed on Qwen2-Audio-7B-Instruct model. We utilize 2 NVIDIA H200 141G GPUs during training. The detailed training hyperparameters have been demonstrated in the Appendix.

## 4.2 RESULTS

We compare our methods with previous methods, including single-modal approaches, conventional multi-modal approaches, and multi-modal LLMs, on both the development set and test set of the DAIC-WoZ database. The detailed comparison results are illustrated in Table 1, Table 2, and Table 3, respectively. Following previous works, we adopt the F1 score for evaluation.

**Evaluation on DAIC-WoZ Dev Set** We present a comprehensive comparison between our proposed multi-modal LLM and previous methods on the DAIC-WoZ development set in Table 1. Additionally, we evaluate the contribution of each individual module in our framework, including the Qwen2-7B model, the Whisper-v3 audio encoder, and a self-supervised vision encoder. Overall, our multi-modal model achieves superior classification performance on the development set of the DAIC-WoZ dataset, consistently outperforming all single-modality baselines.

Text-based models show that Llama2-13B (Touvron et al., 2023; Zhang et al., 2024b) performs best among text-only models, likely due to its larger parameter scale. Among smaller models, Qwen2-7B and Llama2-7B exhibit similar performance but fall short of the 13B variant. Interestingly, GPT-4, despite its scale and zero-shot capabilities, underperforms relative to Llama2-13B. Likewise, RoBERTa surpasses GPT-4 despite its significantly smaller size as well. A similar phenomenon has been observed in Zhang et al. (2024b). This performance gap may be attributed to the nature of depression detection, which emphasizes representation learning over generative modeling, making encoder-based models more suitable.

| Modality | Models | F1 |
|---|---|---|
| Text | RoBERTa 2022 | 0.602 |
| | Llama2-7B 2024b | 0.578 |
| | Llama2-13B 2024b | **0.636** |
| | Qwen2-7B 2024 | 0.564 |
| | GPT4 2024b | 0.571 |
| Audio | HuBERT 2023 | 0.640 |
| | WavLM 2023 | **0.720** |
| | SpeechFormer 2022 | 0.694 |
| | SpeechFormer++ 2023 | 0.709 |
| | Whisper-v3 2023 | 0.694 |
| Video | GSM 2016 | 0.530 |
| | SSL + CLS | **0.668** |
| A+T | AudiBERT 2021 | 0.709 |
| | TOAT 2022 | 0.741 |
| | LSTM 2018 | **0.770** |
| A+T+V | C-CNN 2018 | 0.769 |
| | ConvBiLSTM 2022 | 0.70* |
| | Ours w/o MS | 0.789 |
| | Ours | **0.844** |

Table 1: The performance comparison of our method and other approaches on DAIC-WoZ development set. "*" denotes that the original results are reported with 2 significant digits. "MS" denoting the multi-scale strategy in our inference.

Audio-based models generally outperform text-only models, suggesting that acoustic cues carry richer information for detecting depressive symptoms. In addition, the performance of audio models could benefit from downstream tasks such as speech recognition or emotion recognition (Wu et al., 2023). Notably, WavLM fine-tuned for emotion recognition shows superior performance, surpassing even Whisper-v3-large. This suggests that tasks closely related to depression, such as emotion recognition and ASR, provide transferable knowledge useful for this application.

For video models, our finetuned visual encoder with a classification head achieves the best performance. The main factor that could affect video-based models is the choice of visual feature sets. Since raw videos are not available at the DAIC-WoZ database, only facial feature sets, such as landmarks and action units, are available for depression detection. As the feature set could be rather redundant, the performance of video models could even deteriorate if the feature set selection is inappropriate. Self-supervised pretraining alleviates the issue significantly, as masked autoencoders are designed for images, which possess a redundant nature, and are suitable in our scenario.

Multi-modal approaches that incorporate both audio and text, or integrate all three modalities, generally outperform single-modal baselines. In particular, the inclusion of audio features often leads to significant performance improvements, highlighting the importance of acoustic information in depression detection. Compared with other multi-modal methods, our proposed framework consistently achieves superior results, demonstrating the effectiveness of timestamp-level alignment and the synergy of modality-specific encoders in capturing clinically relevant cues.

**Comparison with Multi-Modal LLMs** Table 2 presents the performance comparison between our method and existing multi-modal LLMs. Together with Table 1, the results demonstrate that incorporating audio significantly enhances the classification performance of LLMs. For instance, augmenting Llama2-13B with acoustic landmarks improves its F1 score from 0.636 to 0.695. A similar trend is observed with Qwen2-7B, where the inclusion of audio elevates the F1 score from 0.578 to 0.720. Our proposed multi-modal framework, which jointly models text, audio, and vi-

sual signals, achieves the highest F1 score of 0.789, validating the benefit of integrating visual cues alongside audio and language inputs. This underscores the advantage of leveraging complementary modalities for capturing the complex and multi-faceted nature of depressive symptoms.

Notably, both our approach and Qwen2-Audio variants outperform LLMs with acoustic landmarks, despite relying on smaller language backbones (7B vs 13B). This suggests that native multi-modal architectures might be more adept at interpreting raw sensory inputs. While acoustic landmarks serve as a lightweight representation of audio, they may omit subtle prosodic or emotional cues that are preserved in the original waveforms. In contrast, models trained end-to-end on raw audio exhibit stronger modality comprehension and more effective feature fusion.

| Model | Base Model | F1 |
|---|---|---|
| Acoustic LLM (Zhang et al., 2024b) | 7B | 0.545 |
| | 7B-Chat | 0.500 |
| | 13B | 0.695 |
| | 13B-Chat | 0.666 |
| Qwen2-Audio (Chu et al., 2024) | 7B | 0.650 |
| | 7B-Instruct | 0.720 |
| Ours w/o audio | 7B | 0.617 |
| | 7B-Instruct | 0.643 |
| Ours | 7B | 0.709 |
| | 7B-Instruct | **0.789** |

Table 2: The performance comparison of our method and multi-modal LLMs on DAIC-WoZ development set. Note that for fair comparison we do not employ model ensemble or multi-scale inference.

**Evaluation on DAIC-WoZ Test Set**    In addition, since the golden labels of the DAIC-WoZ test set have been released, we compare our method with previous state-of-the-art approaches on this benchmark. The quantitative results are presented in Table 3. It can be observed that single-modal approaches yield similar or slightly lower F1 scores on the test set compared to their performance on the development set. In contrast, a recent multi-modal approach that integrates audio, video, and textual information (Jung et al., 2024) achieves significantly better results than single-modal methods. Overall, our method outperforms both previous single-modal and multi-modal approaches on the test set, demonstrating its effectiveness and robustness.

| Models | Modality | F1 |
|---|---|---|
| GloVe-CNN (Campbell et al., 2022) | Text | 0.68* |
| TOAT (Guo et al., 2022) | Audio | 0.647 |
| EmoAudioNet (Othmani et al., 2021) | Audio | 0.66* |
| HiQuE (Jung et al., 2024) | A+T+V | 0.79* |
| Ours | A+T+V | **0.825** |

Table 3: The performance comparison of our method and previous approaches on DAIC-WoZ test set. "*" denotes that the original results are reported with 2 significant digits.

### 4.3 ABLATION STUDIES AND DISCUSSION

In this section, we analyze the source of performance gain in our framework, including the contribution of each modality and the selection of the base model. In addition, we discuss the effectiveness of our proposed timestamp-synchronized data augmentation upon the removal of the interviewer's utterance and context length in subdialogues. The experiments are all conducted on the development set of DAIC-WoZ.

#### 4.3.1 THE CONTRIBUTION OF EACH MODALITY

We further investigate the individual contribution of each modality within our framework. As shown in Table 1, both audio and video modalities enhance depression detection performance. The baseline Qwen2-7B model achieves an F1 score of 0.564 using text alone. Introducing audio features leads to a substantial improvement, raising the F1 score to 0.720. Further incorporation of video features elevates the performance to 0.789. Additionally, our proposed multi-scale sliding-window strategy contributes to model performance significantly, improving the F1 score to 0.844.

An interesting observation is that the addition of audio yields a greater performance gain compared to the inclusion of video, in both instruction-tuned and pre-trained variants. This discrepancy can be attributed to two primary factors. First, as pre-extracted visual features rather than raw video data are utilized in our framework, the model may face information loss, leading to reduced expressive power. Second, our model is fundamentally built upon an audio language modeling architecture. Removing

audio embeddings may disrupt the alignment mechanism across modalities, thereby compromising the model's ability to integrate non-verbal cues effectively.

### 4.3.2 THE CHOICE OF BASE MODEL

Since both pretrained model and instruction-tuned model are available in Qwen2-Audio families, we compare the performance of these two model variants as the base model. The results in Table 2 indicate that the instruction-tuned model provides higher detection performance. The findings in our research are different from previous work (Zhang et al., 2024b), where instruction tuning leads to significant performance deterioration compared with the pretrained model. The reasons for the inconsistency could be the difference in instruction tuning in general LLMs and audio language models. Depression detection involves the analysis of both audio and text; a similar task has been used to finetune the model in instruction tuning. Thus, the instruction-tuned model could be better at the audio analysis task.

### 4.3.3 THE EFFECT OF CONTEXT LENGTH AND INTERVIEWER UTTERANCE REMOVAL

The length of subdialogues plays a crucial role in our framework, as longer contexts generally provide richer cues for depression detection. However, longer subdialogues do not necessarily improve the performance for detection, as the audio records for the interviewer do not contribute to the decision, but even interfere with the depression detection. To address this constraint, we propose to remove the interviewer's utterances during data augmentation, allowing more content from the participant to be retained within the fixed audio window. While this enhances the availability of participant-specific acoustic cues, it also results in the loss of visual information associated with the removed segments. To explore this trade-off, we conduct an ablation study under varying subdialogue lengths, as shown in Figure 4. When the maximum subdialogue length is constrained to 30 seconds, removing the interviewer's speech leads to degraded performance. In this setting, the entire subdialogue can be encoded without truncation, and discarding the interviewer's turns causes unnecessary loss of visual cues, thus impairing multi-modal inference. In contrast, as the subdialogue length increases beyond the model's audio capacity, the removal of interviewer utterances proves beneficial. By prioritizing participant speech within the fixed input window, the model gains access to more relevant acoustic information, leading to improved detection accuracy. However, when the context length becomes excessively long, the performance gain diminishes. This is likely due to reduced dialogue diversity and increased risk of overfitting, as longer subdialogues tend to be less variable.

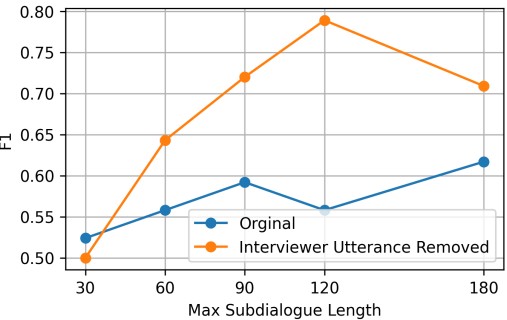

Figure 4: The depression detection performance on subdialogue length with or without interviewer's utterances.

## 5 CONCLUSION

In this study, we propose a multi-modal large language model for depression detection, built upon audio-based language models and augmented with visual understanding capabilities. Experiments on the DAIC-WoZ dataset demonstrate the superiority of our framework over existing multi-modal LLMs. To our knowledge, this is the first work to develop a multi-modal LLM for depression detection that simultaneously integrates textual, audio, and visual modalities. We further provide detailed analyses of how model design and data augmentation strategies affect performance. Overall, our method offers an effective solution for adapting multi-modal LLMs to mental health applications, with potential for broader extension to other domains.

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

## A  IMPLEMENTATION DETAILS

Our framework is implemented using the HuggingFace *transformers* library with PyTorch 2.1. The full hyperparameter configurations used during training are summarized in Table 4. We adopt the AdamW optimizer (Loshchilov & Hutter, 2017) for model optimization. To improve training speed without compromising performance, we enable TensorFloat32 (TF32) computation and apply automatic mixed-precision training using BFloat16 (BF16). For parameter-efficient fine-tuning (PEFT) of the Qwen2-Audio model on the depression detection task, we employ QLoRA (Dettmers et al., 2023), which compresses the base model to 4-bit precision to reduce memory usage and improve computational efficiency. The full training process requires approximately 90+ GPU hours on an NVIDIA H200 141GB GPU. This includes around 40 hours for self-supervised visual pretraining, 20 hours for utterance-level audio-visual alignment, and 30 hours for multimodal instruction tuning. Early stopping is applied in all stages when training loss plateaus.

|  | Stage I | Stage II | Stage III |
| --- | --- | --- | --- |
| Optimizer |  | AdamW |  |
| Learning Rate | 1.5e-4 | 1e-6 | 3e-6 |
| $\beta_1$ |  | 0.9 |  |
| $\beta_2$ | 0.95 | 0.999 |  |
| Weight Decay | 0 | 0.001 |  |
| Batch Size | 128 | 64 | 8 |
| Grad Accum Steps | 8 | 16 | 8 |
| Scheduler |  | Cosine LR |  |
| Num Epochs | 50 | 20 | 3 |
| Warm Up Epochs | 5 | 2 | 0.1 |
| Max Grad Norm | 1.0 | 0.5 |  |
| BF16 |  | True |  |
| TF32 |  | True |  |

Table 4: Training hyperparameters.

## B  DETAILS OF TIMESTAMP-SYNCHRONIZED DATA AUGMENTATION

Following the approach of Wu et al. (2023), we generate subdialogues from the original interview transcripts to mitigate class imbalance and expand the size of the training set. In our data augmentation pipeline, we enforce strict synchronization among transcripts, audio, and video to ensure precise timestamp-level alignment. However, due to varying frame rates across modalities, achieving synchronization presents a technical challenge. For example, audio recordings are typically captured at a 16,000 Hz sampling rate and later converted into Mel-spectrograms with a frame rate of 100 Hz, while video recordings are collected at 30 frames per second (FPS). To address this discrepancy, we constrain the start and end timestamps of each subdialogue to align with whole seconds (i.e., integer-second boundaries).

Additionally, we require each subdialogue to begin with an utterance from the interviewer and conclude with a response from the participant. This design choice ensures that each subdialogue forms

---

**Algorithm 1** Time-Sync Data Augmentation

---

1: $N^+ \leftarrow$ Number of positive samples in the training set
2: $N^- \leftarrow$ Number of negative samples in the training set
3: Set number of subdialogues per positive sample $M^+$
4: Set minimum length of subdialogue in seconds $d_{min}$
5: Set maximum length of subdialogue in seconds $d_{max}$
6: $M^- = N^-/N^+ \times M^+ \leftarrow$ Number of sub-dialogues per negative sample
7: **for** Dialogue $X^{(n)} = (T^n, A^n, V^n), n = 1, 2, ..., N$ **do**
8:     $D \leftarrow$ Dialogue length in seconds
9:     $\{\varepsilon_i\} \leftarrow$ Interviewer utterance start timestamps
10:     $\{\varepsilon_p\} \leftarrow$ Participant utterance end timestamps
11:     **if** $X^{(n)}$ is positive **then**
12:         $M \leftarrow M^+$
13:     **else**
14:         $M \leftarrow M^-$
15:     **end if**
16:     **for** Sub-dialogue $X^{(n)m}, m = 1$ to $M$ **do**
17:         Sample length $d$ uniformly from $(d_{min}, d_{max})$
18:         Sample start timestamp $\varepsilon'_s \in \{\varepsilon_i\}$ from range $(0, D - d)$
19:         Round the start timestamp to its closet integer second $\varepsilon_s \leftarrow \lfloor \varepsilon'_s \rfloor$
20:         $\varepsilon_{tmp} = \varepsilon_s + d \leftarrow$ Raw end timestamp
21:         Sample end timestamp $\varepsilon'_e \in \{\varepsilon_p\}$ and $\min |\varepsilon'_e - \varepsilon_{tmp}|$
22:         Round the end timestamp to its closet integer second $\varepsilon_e \leftarrow \lceil \varepsilon'_e \rceil$
23:         Generate subdialogue $T^{(n)m} \leftarrow T^{(n)}_{\varepsilon_s:\varepsilon_e}$
24:         Obtain the raw audio segment $A'^{(n)m} \leftarrow A^{(n)}_{\varepsilon_s:\varepsilon_e}$
25:         Obtain the raw visual segment $V'^{(n)m} \leftarrow V^{(n)}_{\varepsilon_s:\varepsilon_e}$
26:         Remove the interviewer utterances $A^{(n)m} \leftarrow A'^{(n)m}$ and $V^{(n)m} \leftarrow V'^{(n)m}$
27:         Subdialogue $X^{(n)m} = (T^{(n)m}, A^{(n)m}, V^{(n)m})$
28:     **end for**
29: **end for**

---

a complete and contextually coherent conversational unit, with a clear initiation and response structure. Such a constraint preserves the semantic continuity and logical flow within each segment, making them more suitable for downstream tasks that rely on natural discourse patterns. Moreover, this structure aligns with the training paradigm of large language models, which are typically pre-trained on large-scale dialogue corpora. By maintaining this dialogue consistency, we enhance the model's ability to interpret the subdialogues effectively within a familiar conversational framework.

## C   THE PROMPT DESIGN FOR INSTRUCTION TUNING

During instruction tuning, we design a system prompt to guide the behavior of the language model. Given that the Qwen2-Audio-Instruct model has been fine-tuned on audio analysis tasks, we adopt a chat-based prompt template to elicit model responses. Notably, we use the same prompt design for both the Qwen2-Audio family and our multi-modal LLM. This consistency is based on our integration strategy, where visual embeddings are directly added to the audio embeddings without modifying the model architecture. Therefore, we assume that the model can still function effectively even without explicitly referencing visual information in the prompt.

**System Prompt** `Below is a conversation between an interviewer and a participant. Please analyze the transcripts and audio, and find whether the participant is affected by depression.`

**Instructions** `Audio: {audio} \n Interview conversation: {transcripts} \n Response: \n`

