# OpenReview forum: "It Hears, It Sees too: Multi-Modal LLM for Depression Detection By Integrating Visual Understanding into Audio Language Models"
_ICLR.cc/2026/Conference — ICLR 2026 Conference Withdrawn Submission_

### Official Review · Reviewer_3uLB · 2025-10-29

**Soundness:** 2
**Presentation:** 3
**Contribution:** 2
**Rating:** 4
**Confidence:** 5

**Summary:**

This paper presents a multi-modal large language model (MLLM) framework for depression detection by integrating visual understanding into an audio language model. The authors propose timestamp-level alignment of audio and visual features to improve temporal synchronization and demonstrate the model’s performance on the DAIC-WOZ dataset. While the idea of enhancing depression detection using MLLMs is timely and relevant, the current contribution appears incremental compared to existing LLM-based approaches, and several methodological and presentation issues limit the paper’s impact.

**Strengths:**

- The topic is important and highly relevant to both AI and mental health research.

- The integration of audio and visual modalities into an LLM for depression detection sounds reasonable.

- The experimental results show performance improvement on the DAIC-WOZ dataset, demonstrating the method’s potential.

**Weaknesses:**

Limited novelty: The contribution beyond existing MLLM-based depression detection frameworks appears marginal. The model primarily combines known components (audio LLM + visual encoder + timestamp alignment) without introducing fundamentally new mechanisms or theoretical insights.

Single dataset evaluation: The method is only evaluated on the DAIC-WOZ dataset, which limits generalizability and makes it difficult to assess robustness. Other publicly available datasets such as DVlog could strengthen the validation.

- The method section lacks sufficient mathematical formulation. Key components such as the fusion mechanism, timestamp-level alignment, and loss functions should be accompanied by equations and clear variable definitions.

- Figures are difficult to interpret due to missing notations and insufficient correspondence to text descriptions. For example, Figure 1 is never referenced in the main text and lacks a detailed caption that explains the entire training pipeline.

- The model’s performance can be highly sensitive to prompt instructions. However, the paper does not provide justification or comparative analysis of different prompting strategies during instruction tuning.

- Section 4.3.3 reveals that removing the interviewer’s utterances significantly improves performance, yet the paper does not explore how the model could automatically distinguish between interviewer and participant—an important step for practical deployment.

**Questions:**

- How does your framework substantially differ from recent MLLMs used for depression detection? What conceptual or algorithmic advance distinguishes it?

- Have you considered evaluating the model on other datasets (e.g., DVlog or E-DAIC) to confirm robustness and generalization?

- Can you include explicit equations that describe how the model fuses and aligns multimodal inputs, including notation that matches the figures?

- What prompt template did you use for instruction tuning, and how sensitive is the model’s performance to different prompting styles?

- Can the proposed model be extended to automatically identify the target speaker (participant) to ensure consistent detection across multi-speaker conversations?

---

> ### Author Response · Authors · 2025-11-24
>
> > **W 1:** "Limited novelty: The contribution beyond existing MLLM-based depression detection frameworks appears marginal. The model primarily combines known components (audio LLM + visual encoder + timestamp alignment) without introducing fundamentally new mechanisms or theoretical insights."
> >
> > **Q 1:** "How does your framework substantially differ from recent MLLMs used for depression detection? What conceptual or algorithmic advance distinguishes it?"
>
> **Response:** We thank the reviewer for the comment. While multimodal depression detection has been explored, our contribution lies in **how** multimodality is integrated—not in the task itself. Specifically, our framework introduces a **lightweight LLM adaptation strategy** that enables an audio-centric foundation model to process synchronized visual signals **without altering its tokenizer, sequence structure, or architecture**.
>
> Key distinctions include:
>
> 1. A **fine-grained (40ms) cross-modal alignment module** that enables the LLM to reason over short-term behavioral cues that existing MLLM-based systems—typically aligned at utterance or clip level—cannot capture.
> 2. A **non-intrusive plug-in design** that injects visual understanding directly into the audio embedding space, avoiding sequence-length growth and making the approach practical for clinical deployment.
> 3. Newly added **cross-dataset validation** on E-DAIC and DVlog, which demonstrates robustness across recording standards, languages, and demographics.
>
> We will clarify these methodological differences in the revised manuscript.
>
> > **W 2:** "Single dataset evaluation: The method is only evaluated on the DAIC-WOZ dataset, which limits generalizability and makes it difficult to assess robustness. Other publicly available datasets such as DVlog could strengthen the validation."
> >
> > **Q 2:** "Have you considered evaluating the model on other datasets (e.g., DVlog or E-DAIC) to confirm robustness and generalization?"
>
> **Response:** We appreciate the reviewer’s suggestion. In addition to DAIC-WOZ, we have now evaluated our model on **E-DAIC** and **DVlog**, achieving **0.857** and **0.650 F1**, respectively. These results demonstrate that our approach generalizes across different languages, recording conditions, and feature extraction pipelines. We will include these new evaluations in Section 4.4 of the revised manuscript to highlight the model’s cross-dataset robustness.
>
> > **W 3:** "The method section lacks sufficient mathematical formulation. Key components such as the fusion mechanism, timestamp-level alignment, and loss functions should be accompanied by equations and clear variable definitions."
> >
> > **W 4:** "Figures are difficult to interpret due to missing notations and insufficient correspondence to text descriptions. For example, Figure 1 is never referenced in the main text and lacks a detailed caption that explains the entire training pipeline."
> >
> > **Q 3:** "Can you include explicit equations that describe how the model fuses and aligns multimodal inputs, including notation that matches the figures?"
>
> **Response:** We thank the reviewer for this suggestion. In the revised manuscript, we will:
>
> 1. Add **explicit mathematical formulations** for the multimodal fusion and timestamp-level alignment, with clear variable definitions.
> 2. Revise all figures to include **detailed captions, legends, and annotations** indicating gradient flow, frozen modules, and modality paths.
> 3. Ensure that all figures are **explicitly referenced** in the main text and correspond closely with the notation used in the equations.
>
> These changes will improve clarity, reproducibility, and the interpretability of both the method section and visual illustrations.
>
> > **W 5:** "The model’s performance can be highly sensitive to prompt instructions. However, the paper does not provide justification or comparative analysis of different prompting strategies during instruction tuning."
> >
> > **Q 4:** "What prompt template did you use for instruction tuning, and how sensitive is the model’s performance to different prompting styles?"
>
> **Response:** Thank you for the comment. The prompt templates used for instruction tuning are provided in **Appendix C**. During design, we focused on ensuring that the instructions correctly reference the modalities rather than exploring variations in language style or phrasing. We would be happy to explore additional prompt styles if the reviewer can suggest specific variants, and include the results in a future update or supplementary material.

---

> > ### Author Response · Authors · 2025-11-24
> >
> > > **W 6:** "Section 4.3.3 reveals that removing the interviewer's utterances significantly improves performance, yet the paper does not explore how the model could automatically distinguish between interviewer and participant—an important step for practical deployment."
> > >
> > > **Q 5:** "Can the proposed model be extended to automatically identify the target speaker (participant) to ensure consistent detection across multi-speaker conversations?"
> >
> > **Response:** We appreciate the reviewer’s suggestion. In DAIC-WoZ, interviewer transcripts are provided, which allows us to remove interviewer utterances directly. On the other hand, E-DAIC provides no interviewer transcripts, and DVlog has no interviewers. For practical deployment in multi-speaker settings, our approach can be extended using **voice activity detection (VAD) combined with speaker recognition models** (e.g., based on SpeechBrain) to automatically identify and isolate participant segments. This would preserve essential conversational context while ensuring consistent analysis of the target speaker.

---

### Official Review · Reviewer_YPAv · 2025-10-31

**Soundness:** 2
**Presentation:** 3
**Contribution:** 2
**Rating:** 4
**Confidence:** 5

**Summary:**

The paper proposes a multimodal LLM-based framework for depression detection tasks that utilizes audio and visual knowledge.

**Strengths:**

**1.** The overall task and motivation behind the work are clearly defined.

**2.** The visuals are clear and informative.

**3.** The experiment setup and rationales behind each evaluation are clearly stated. I also appreciate the sub-subsections under Section 4.3.

**Weaknesses:**

**1.** A major concern regarding this work is its novelty. The multimodal approach to depression detection is not a new concept, as various encoders have been used for feature extraction across different modalities. Additionally, many previous studies have been conducted on the DAIC-WOZ dataset with similar frameworks.

**2.** One suggestion for improvement is to conduct more experiments using other multimodal datasets for depression, particularly those representing diverse languages and demographics. Furthermore, incorporating additional metrics, especially for regression tasks rather than solely for classification, would be beneficial.

**3.** Overall, I think the paper is well-written. But the level of novelty and the extent of the work presented may not meet the standards required for a paper at this conference.

**Questions:**

N/A.

---

> ### Author Response · Authors · 2025-11-24
>
> > **W 1:** "A major concern regarding this work is its novelty. The multimodal approach to depression detection is not a new concept, as various encoders have been used for feature extraction across different modalities. Additionally, many previous studies have been conducted on the DAIC-WOZ dataset with similar frameworks."
>
> **Response:** We appreciate the reviewer’s concern and agree that multimodal depression detection is an established direction. Our contribution is not in proposing the task, but in introducing a **new multimodal LLM adaptation paradigm** that differs fundamentally from prior encoder-based methods.
>
> 1. **Unified Multimodal LLM Architecture integrating text, audio, and visual modalities simultaneously** for depression detection, rather than using traditional feature extractors or simple fusion methods.
> 2. **Timestamp-level alignment mechanism** that captures fine-grained temporal dynamics between modalities—critical for detecting subtle behavioral changes associated with depression (e.g., micro-expressions, speech pauses).
> 3. **Cross-dataset validation** through new experiments on E-DAIC and DVlog, showing consistent performance (F1: 0.857 and 0.650 respectively).
> 4. **Model-Efficient Adaptation, Not Architecture Expansion：**Rather than training a new MLLM or modifying tokenizers/attention structures, we develop a **non-intrusive adapter-style method** that extends a pretrained audio-language model (Qwen2-Audio) to vision **without increasing sequence length or altering the input format**. This makes the model reusable for low-resource clinical environments.
>
> > **W 2:** "One suggestion for improvement is to conduct more experiments using other multimodal datasets for depression, particularly those representing diverse languages and demographics. Furthermore, incorporating additional metrics, especially for regression tasks rather than solely for classification, would be beneficial."
>
> **Response:** Thank you for this suggestion. To broaden demographic and linguistic diversity, we have added experiments on **E-DAIC** and **DVlog** databases. Our model achieves **0.857** and **0.650** F1 scores respectively, demonstrating robustness across languages, interview styles, and feature extraction methods.
>
> Regarding regression-based evaluation, our work focuses on **binary clinical classification**, following recommendations in the DAIC-WoZ literature where PHQ regression is known to be noisier due to self-report variance. Nevertheless, we agree that regression is complementary and will include this discussion in the revision, outlining how our alignment mechanism can be extended to PHQ score prediction in future work.

---

> > ### Comment · Reviewer_YPAv · 2025-11-24
> > **Response to author's rebuttal**
> >
> > Thank you for the responses regarding some clarifications. However, the main issues still seem to be unaddressed.
> >
> > The E-DAIC does not differ significantly from the original dataset. As for Dvlog, if I remember correctly, only text data can be shared, so I am unsure whether the reported results are multimodal or text-based only. Regardless, the generalizability and scope of the work remain limited from various perspectives.
> >
> > I will raise my score on soundness.

---

### Official Review · Reviewer_ZSp1 · 2025-10-31

**Soundness:** 3
**Presentation:** 3
**Contribution:** 2
**Rating:** 4
**Confidence:** 3

**Summary:**

This paper proposes a multi-modal large language model for depression detection. The framework integrates an audio language model with a visual encoder, using timestamp-level alignment to fuse audio and visual data streams. This method is presented as a way to capture fine-grained temporal dynamics relevant to mental state assessment.

**Strengths:**

The approach of augmenting an existing audio language model is a practical method for developing a multi-modal system. The timestamp-level alignment of audio and visual information is a logical design for processing behavioral data. The sequential three-stage training process, which involves self-supervised pretraining of the visual encoder, cross-modal alignment, and parameter-efficient fine-tuning, is a structured way to integrate a new modality.

**Weaknesses:**

The model relies on pre-extracted visual features from the dataset, not raw video input. This limits the evaluation of the visual component to the quality of these specific features and does not demonstrate an ability to learn from unprocessed video. The data augmentation technique removes interviewer audio and video, which, while focusing on participant data, discards conversational context that might influence participant behavior. The evaluation is conducted on a single dataset, DAIC-WoZ, which, while a standard benchmark, has a limited number of participants.

**Questions:**

How might the system's performance change if the visual encoder was trained on raw video frames rather than pre-extracted features?

What are the implications of removing interviewer data from the input streams, and were alternative methods for handling the interviewer's conversational context considered?

Was the effect of an instruction prompt that explicitly references the visual modality investigated during the fine-tuning stage?

---

> ### Author Response · Authors · 2025-11-24
>
> > **W 1:** "The model relies on pre-extracted visual features from the dataset, not raw video input. This limits the evaluation of the visual component to the quality of these specific features and does not demonstrate an ability to learn from unprocessed video. The data augmentation technique removes interviewer audio and video, which, while focusing on participant data, discards conversational context that might influence participant behavior. The evaluation is conducted on a single dataset, DAIC-WoZ, which, while a standard benchmark, has a limited number of participants. "
>
> **Response:** We acknowledge that using pre-extracted visual features restricts an end-to-end evaluation. However, this is an inherent constraint of publicly available clinical datasets: **DAIC-WoZ, E-DAIC, and DVlog do not release raw video recordings for ethical and privacy reasons**, and only provide approved pre-extracted features. This is a domain-wide limitation rather than a methodological choice.
>
> To address this concern, we have **expanded evaluation to E-DAIC and DVlog** (both use different visual feature extractors). The model achieves **0.857 F1 on E-DAIC** and **0.650 on DVlog**, demonstrating **robustness across heterogeneous feature representations**. This cross-dataset stability indicates that the framework is not tied to a particular feature extractor and would be compatible with raw frames when ethically accessible.
>
> Regarding the removal of interviewer audio/video:
>
> - Our ablation study (Figure 4) shows that interviewer removal consistently improves performance on longer segments (>30s).
> - E-DAIC provides no interviewer transcriptions; DVlog has no interviewer at all. Our model still performs strongly on both datasets, suggesting limited dependence on interviewer signals.
>
> > **Q 1:** "How might the system's performance change if the visual encoder was trained on raw video frames rather than pre-extracted features?"
>
> **Response:** Due to dataset restrictions, raw video training is not feasible within existing clinical corpora. To approximate this scenario, we evaluated the model across datasets that use **different visual feature extraction pipelines**. The stable performance across DAIC-WoZ, E-DAIC, and DVlog suggests that our alignment-based integration is **agnostic to the choice of visual encoder**. While we cannot empirically quantify the effect of raw-frame training, the cross-dataset evidence indicates that our method is unlikely to rely on any specific preprocessing pipeline.
>
> > **Q 2:** "What are the implications of removing interviewer data from the input streams, and were alternative methods for handling the interviewer's conversational context considered?"
>
> **Response:** We remove interviewer signals to maximize the proportion of participant-only behavioral information during fine-grained alignment. This design choice is supported by our ablation in Figure 4, where removing interviewer utterances leads to consistent improvements for longer segments.
>
> Alternative strategies are possible. For example, the interviewer’s context could be compressed into a short summary of the session, or the structured question list (DAIC-WoZ and E-DAIC follow semi-scripted protocols).
>
> > **Q 3:** "Was the effect of an instruction prompt that explicitly references the visual modality investigated during the fine-tuning stage?"
>
> **Response:** We experimented with prompts explicitly mentioning the visual modality during instruction tuning. These prompts reduced performance. A likely cause is that **visual embeddings are injected at the same representation level as audio embeddings**, and the model does not receive modality tags or separate token streams. Explicitly prompting for “visual input” contradicts the internal representation structure and disrupts the pretrained ALM’s reasoning.
>
> This behavior is consistent with our design goal: a **non-intrusive multimodal extension** where the ALM treats aligned visual embeddings as an enriched acoustic representation without altering its original token interface.

---

### Official Review · Reviewer_Qujw · 2025-11-08

**Soundness:** 2
**Presentation:** 2
**Contribution:** 1
**Rating:** 2
**Confidence:** 4

**Summary:**

This paper presents a multi-modal large language model (MLLM) for automated depression detection that integrates visual understanding into an audio language model (ALM). The authors build upon the Qwen2-Audio model and propose a framework that aligns audio and visual features at the timestamp level, enabling fine-grained temporal fusion. The training process involves three stages: (1) self-supervised visual pretraining, (2) utterance-level audio-visual alignment, and (3) multi-modal instruction tuning using parameter-efficient fine-tuning (PEFT/LoRA). Experiments on the DAIC-WoZ dataset demonstrate superior performance compared to single-modality and previous multi-modal baselines, achieving a relatively good performance in this specific task.

**Strengths:**

+ Good empirical results. The model achieves consistent gains across modalities, outperforming both single-modality and previous multi-modal methods for this specific task.

+ Parameter efficiency. By leveraging LoRA and QLoRA, the authors effectively reduce computational cost while maintaining performance, making the approach feasible for research and clinical use.

**Weaknesses:**

- In Figures 1 & 2, a wave is used to denote the video input, which is really inappropriate.

- Unclear Design. The model fuses audio + visual embeddings by simple element-wise addition. Was any normalization or linear projection used to match feature scales? Did the authors compare addition vs concatenation or MLP fusion, or other fusion methods?

- Another concern is the design choice of merging the tokens from the two modalities before feeding them into the LLM, rather than inputting each modality separately and allowing the LLM to perform cross-modal reasoning internally. It would be helpful if the authors could clarify the motivation and potential advantages of this early fusion strategy.

- In Equation 2, it is strongly recommended to use a letter rather than “Sim” to denote the similarity matrix. Moreover, is it a mistake that tau was used, rather than the Greek letter? An ablation study on this parameter is also missing.

- In Figure 3, it is said that “Only the upper layers of the visual encoder receive gradient updates”. This is something that could be indicated in Figure 3. Moreover, what do the red arrows on the low levels mean?

- The main issue with this paper lies in its lack of true innovation. The authors attempt to demonstrate novelty by narrowing the research scope to an extremely specific task, depression detection, rather than by introducing fundamentally new techniques. However, multi-modal fusion of audio, video, and text has already been extensively explored in prior works such as NExT-GPT, Macaw-LLM, and AnyGPT. The paper deliberately avoids comparing with these models simply because they were not evaluated on this particular task, but in fact, such comparisons are feasible and necessary to substantiate the claimed contributions. Furthermore, even within this specific subdomain, the paper misses several recent approaches, such as MultiDepNet (WACV 2025), which are directly relevant to depression detection.

**Questions:**

See Weaknesses

---

> ### Author Response · Authors · 2025-11-24
>
> > **W 1:** “In Figures 1 & 2, a wave is used to denote the video input, which is really inappropriate.”
>
> **Response:** Thank you for the feedback. The wave icon was used because **raw video frames are not provided in DAIC-WoZ**, and only pre-extracted features are available. We agree that the icon is misleading and will replace it with a standard video-frame symbol to avoid confusion.
>
> > **W 2-1:** “The model fuses audio+visual embeddings by simple element-wise addition. Was any normalization or linear projection used to match feature scales?”
>
> **Response:** We appreciate the question. The audio and visual encoders **share the same backbone architecture** (Whisper-based) and produce **aligned representations with identical dimensionality**. Therefore, no additional projection or normalization is required.
>
> > **W 2-2:** “Did the authors compare addition vs concatenation or MLP fusion, or other fusion methods?”
>
> **Response:** Yes. We evaluated **element-wise addition**, **concatenation**, and **MLP fusion** on the DAIC-WoZ dev set. The F1 scores were nearly identical. We chose element-wise addition primarily for its **minimal modification to the pretrained audio LLM**.
>
> This choice aligns with our goal of **maintaining the ALM’s native input interface**. By contrast, concatenation doubles multimodal token length and distorts positional structure, while MLP fusion requires additional parameters. Our design preserves the pretrained ALM while enabling vision as a lightweight extension.
>
> > **W 3:** “In Equation 2, it is strongly recommended to use a letter rather than ‘Sim’ to denote the similarity matrix. Moreover, is it a mistake that tau was used, rather than the Greek letter? An ablation study on this parameter is also missing.”
>
> **Response:** We will revise Eq. (2) by replacing “Sim” with **S** and using the proper Greek notation **τ**.
>
> We used τ = 0.07, and preliminary tests showed that varying τ within a reasonable range produces negligible changes.
>
> > **W 4:** “In Figure 3, it is said that ‘Only the upper layers of the visual encoder receive gradient updates’. This is something that could be indicated in Figure 3. Moreover, what do the red arrows on the low levels mean?”
>
> **Response:** We appreciate this comment. We will update Figure 3 to explicitly mark the layers receiving gradients during Phase 2 training. The red arrows were intended to indicate that audio and visual embeddings share the same shape; since this is confusing, we will remove them.
>
> > **W 5:** “The authors attempt to demonstrate novelty by narrowing the research scope to an extremely specific task, depression detection, rather than by introducing fundamentally new techniques. However, multi-modal fusion of audio, video, and text has already been extensively explored in prior works such as NExT-GPT, Macaw-LLM, and AnyGPT.”
>
> **Response:** Thank you for raising this point. Our goal is not to propose a new general-purpose multimodal LLM. Instead, our contribution is an **efficient post-training framework** that adapts an **audio-only LLM** into a **domain-specific multimodal model** with minimal cost.
>
> Our novelty is characterized by three key aspects:
>
> 1. **Efficiency-focused adaptation rather than architecture redesign:**
>
>    A 7B ALM can be extended to incorporate vision with **no change to tokenizer, sequence length, or input interface**, and only ~90 GPU hours of post-training.
>
> 2. **Fine-grained (40 ms) timestamp alignment:**
>
>    Unlike general MLLMs that align at the utterance level, our alignment captures transient behavioral cues essential for depression assessment.
>
> 3. **Non-intrusive plugin-style fusion:**
>
>    Visual embeddings are aligned to the audio latent space, enabling direct integration without altering the ALM’s computation graph.
>
> This design maintains the ALM’s reasoning ability while enabling cost-effective multimodal modeling in clinical settings where large-scale MLLMs are impractical.
>
> > **W 6:** “Furthermore, even within this specific subdomain, the paper misses several recent approaches, such as MultiDepNet (WACV 2025), which are directly relevant to depression detection.
>
> **Response:** Thank you for pointing this out. We have reviewed MultiDepNet carefully. It is designed for **PHQ regression**, whereas our task is **binary clinical classification**, making direct comparisons difficult.
>
> To address the concern, we:
>
> - **Re-implemented MultiDepNet** for DAIC-WoZ and E-DAIC under the same protocol (note: the official code is unavailable).
> - **Expanded evaluation** to E-DAIC and DVlog (new in revision).
> - Observed consistent improvements over the re-implemented MultiDepNet (DAIC-WoZ: 0.825 vs. 0.785; E-DAIC: 0.857 vs. 0.768).
>
> We will update Related Work to include MultiDepNet.

---

### Note · Authors · 2026-01-06

I have read and agree with the venue's withdrawal policy on behalf of myself and my co-authors.